# SPARSE SPATIO TEMPORAL RECONSTRUCTION WITH CLOSABLE KERNEL SPACE

## ABSTRACT

Quantifying spatio-temporal (ST) measures of dynamical systems is a crucial problem with wide ranging applications in climate modeling, epidemiology, physical processes to name a few. We are interested in the same but motivated by a rather practical scenario where sparse information is collected non-uniformly. To reconstruct the underlying dynamical system under such constraints, we propose a novel algorithm for learning the Koopman operator via a Reproducing Kernel Hilbert Space (RKHS) based on the *Laplacian Kernel Extended Dynamic Mode Decomposition* (Lap-KeDMD). We further show that our kernel space resolves a fundamental issue that is required for a faithful reconstruction of the Koopman operator of the underlying ST data by proving its closability. We demonstrate our method on standard benchmark cases – Burger's Equation, fluid flow across cylinder and Duffing Oscillator. We then reconstruct the Koopman operator for a real ST Seattle traffic flow data that is collected non-uniformly. Necessary comparisons are made between the current state of the art kernel methods corresponding to Gaussian Radial Basis Function (GRBF) Kernel. Such empirical comparisons leads us to conclude that Lap-KeDMD remarkably outperforms as compared to that of aforementioned counter-part thereby, making the Laplacian Kernel a robust choice for such ST quantification.

## 1 INTRODUCTION

Spatio-temporal (ST) data have become ubiquitous in the current age where surveillance systems and sensors are recording large volumes of video data in airports, roads, weather monitoring etc. One important aspect of dealing with ST data is the task of prediction. Numerous applications in traffic prediction, video action recognition, anomaly detection have led to advancement in deep learning techniques like spatio-temporal convolution, spatio-temporal transformers, graph based spatio-temporal transformers to name a few (see, e.g. He et al. (2019)Zhou et al. (2016)Grigsby et al. (2021)Yu et al. (2020)Yan et al. (2021)Li et al. (2022).

On the other hand, there is an increased focus on understanding ST data from a dynamical systems perspective (see Budišić et al. (2012)) where the task is not ST prediction but rather *spatio-temporal reconstruction*. Theoretical investigations of dynamical systems, first initiated by Poincare, has recently made huge advancements due to many applications arising in system design of engineered systems, optimization and control and understandings of complex physical phenomena like turbulence. Currently, data-driven methods provide a promising route in ST reconstruction by learning *spatio-temporal modes* of the dynamical system. One of the key and useful data-driven algorithm to provide ST modes is *Dynamic Mode Decomposition* developed by Schmid (2010). DMD in principle is an unsupervised machine learning (ML) algorithm Fujii & Kawahara (2019), based on *Krylov subspaces* Saad (2003) and *Arnoldi algorithm* Arnoldi (1951).

As far as ST reconstruction in concerned, ML architectures for instance Rubanova et al. (2019) (ODE-RNNs) or Ye et al. (2020); Murata et al. (2020); Hasegawa et al. (2020), (Brunton & Kutz, 2022, Chapter 6 Section 8, Page 236) based on Convolutional Neural Networks and Autoencoders offer a competitive platform to simulate, model and forecast complex, chaotic non-linear dynamical systems but face serious issues in analyzing the underlying dynamics as they lack tendency to operate or evolve with respect to *time* (cf. Mezić (2021); Sarker (2021); Haggerty et al. (2023)) Interestingly, such pitfall is overcome up by the data-driven techniques powered by the spectral analysis

of the infinite-dimensional linear Koopman operators (cf. Koopman (1931); Koopman & Neumann (1932); Mezić (2005)) over the underlying Hilbert space Williams et al. (2015a); Li et al. (2017) and such framework of DMD is referred as *Extended DMD* (eDMD). However, since such data driven methods are leveraged by the finite rank representations of the Koopman operators, which fails to be compact Singh & Kumar (1979), hence making the usual $L^2-$ Hilbert space incomprehensible.

In such cases, choosing (adequate) RKHSs for the action of Koopman operators to perform DMD naturally becomes a more justifiable options, which is called as *Kernel Extended DMD* (KeDMD) Baddoo et al. (2022); Klus et al. (2018; 2020); Rosenfeld et al. (2022) and is central to this paper. Recent investigation in KeDMD by Colbrook in (Colbrook, 2023, Page 44) recommends that one should choose the kernel function space so that the Koopman operator on the underlying RKHS is not just densely defined but also '*closable*'. However, coming up with such kernel spaces is non-trivial Ikeda et al. (2022b;a). Among the popular choices of the kernel functions, is GRBF Kernel (Steinwart et al., 2006, Page 4639) which not only just finds application in data-driven modeling but also in various ML platforms such as support vector machines and etc.

There is a growing interest of data predictions based on collecting ST-reconstructions via Koopman operators over the underlying RKHS. However, the problem of ST-reconstruction for data predictions while collecting snapshots irregularly is still under-investigated. Ideally, *full dataset* is needed to initiate DMD and is synthesis by stacking the data snapshots collected in a regular time-sampled manner. Experimental analysis in Bevanda et al. (2024); Tu et al. (2014); Klus et al. (2016); Rosenfeld et al. (2022); Brunton & Kutz (2022); Williams et al. (2015a); Schmid (2010; 2022); Baddoo et al. (2022) (and references there in) seems to enjoy such hypothetical-ideal situation, which seems to be quite far from the actual reality. Nevertheless, describing and forecasting the time evolution of dynamical systems still remains a challenging problem in the setting of limited irregularly placed data snapshots. Lack of investigation in this direction motivates this very paper, which provide the solution to this problem by leveraging Kernelized eDMD and novelty of Laplacian Kernel.

Recovering ST modes is essentially a key to the data-driven methods and fueled by this, this paper aims to provide the solution to the challenge of recovering data dynamics when we have irregular and sparse data in conjunction with limited dataset.

Our main contribution in the proposed work is summarized below:

1. We develop an RKHS by the Laplacian Kernel embedded as $L^2-$Lebesgue measure in Subsubsection 3.1 and provide the Koopman operator theoretic quantification (cf. Theorem A.12 and Theorem A.19) of Koopman operators over this RKHS.

2. We show that the Koopman operators over this RKHS space is *closable*(Theorem 3.2) while showing that the current GRBF Kernel leads to a failure of closability. (Theorem 3.3).

3. We develop an algorithm to identify the Koopman operator over this RKHS based on irregular and sparsely sampled data and tabulate our results in Subsubsection 3.2. In particular, we showcase our results on real ST data collected sparsely from speed sensors in Seattle.

## 2 MATHEMATICAL BACKGROUND ON DYNAMICAL SYSTEM, KOOPMAN OPERATORS AND KERNEL SPACES

In reconstruction of ST data, we have Sampling-flow assumption that there is an underlying dynamical system and the data snapshots are a noisy observable of that dynamical system collected in certain sense. This particular assumption is common and standard to initiate the data-driven methods discussion and can be learned from Bevanda et al. (2024); Brunton & Kutz (2022); Colbrook (2023); Giannakis & Das (2020) with references present there in as well.

**Assumption 2.1** (Sampling-flow assumption). *Let $\mathcal{M}$ be a metric space and $\mathbf{F}_t : \mathcal{M} \to \mathcal{M}$ be the flow as defined in equation 1 along with the Borel-probability measure $\mu$ whose support $\operatorname{supp} \mu = X$. Let the system be sampled at a fixed-time-instant, say $\Delta t \, (> 0)$ such that $\mathbf{F}_{n\Delta t} : \mathcal{M} \to \mathcal{M}$.*

$$\frac{d}{dt}\boldsymbol{x}(t) = \mathbf{f}\left(\boldsymbol{x}(t)\right) \implies \mathbf{F}_t\left(\boldsymbol{x}(t_0)\right) = \boldsymbol{x}\left(t_0\right) + \int_{t_0}^{t_0+t} \mathbf{f}\left(\boldsymbol{x}(\mathtt{t})\right) d\mathtt{t}. \tag{1}$$

**Definition 2.1** (Koopman Operators). *Under Assumption 2.1, the dynamical flow $\mathbf{F}_t$ induces a linear map $\mathcal{K}_{\mathbf{F}_t}$ on the vector space of complex-valued functions on $\mathcal{M}$ and on $X$ defined as*

$$\mathcal{K}_{\mathbf{F}_t} : L^2(\mu) \to L^2(\mu) \implies \mathcal{K}_{\mathbf{F}_t} g := g \circ \mathbf{F}_t. \tag{2}$$

We now provide definition of Koopman eigenfunction $\Phi_\lambda$ corresponding to eigenvalue $\lambda \in \mathbb{C}$ Mezić (2020).

**Definition 2.2.** *Koopman eigenfunction $\Phi_\lambda \in C(\mathbb{X})$ satisfies $\Phi_\lambda(\boldsymbol{x}) = \exp(-\lambda t)\Phi_\lambda(\mathbf{F}_t(\boldsymbol{x}))$ over $t \in [0, T]$.*

The sampling flow assumption give rise to the discrete dynamical system $(n, \mathcal{M}, \mathbf{F})$ based on which one can discuss the interaction of Koopman operator equation 4 and the snapshots of dynamic flow equation 3; these are respectively given as follows

**Definition 2.3.** *Consider $(n, \mathcal{M}, \mathbf{F})$ be the discrete dynamical system where $n \in \mathbb{Z}$ is time $\mathcal{M} \subseteq \mathbb{R}^n$ is the state space and $\boldsymbol{x} \mapsto \mathbf{F}(\boldsymbol{x})$ is the dynamics. Then the data-set of snapshots of pairs corresponding to the discrete dynamical system $(n, \mathcal{M}, \mathbf{F})$ is given as following:*

$$\underbrace{\begin{bmatrix} | & | & | & | \\ \boldsymbol{x}_1 & \boldsymbol{x}_2 & \cdots & \boldsymbol{x}_m \\ | & | & | & | \end{bmatrix}}_{\boldsymbol{X}} \overset{\boldsymbol{x}_i \mapsto \mathbf{F}(\boldsymbol{x}_i)}{\mapsto} \underbrace{\begin{bmatrix} | & | & | & | \\ \mathbf{F}(\boldsymbol{x}_1) & \mathbf{F}(\boldsymbol{x}_2) & \cdots & \mathbf{F}(\boldsymbol{x}_m) \\ | & | & | & | \end{bmatrix}}_{\boldsymbol{X}^\bowtie} \overset{\boldsymbol{y}_i = \mathbf{F}(\boldsymbol{x}_i)}{:=} \begin{bmatrix} | & | & | & | \\ \boldsymbol{y}_1 & \boldsymbol{y}_2 & \cdots & \boldsymbol{y}_m \\ | & | & | & | \end{bmatrix}.$$

$$\tag{3}$$

With a slight abuse of notation to the Koopman operator as $\mathcal{K}$ (instead of $\mathcal{K}_{\mathbf{F}_t}$ in equation 2), we understand that the Koopman operator acting on the observable $\phi : \mathcal{M} \subset \mathbb{C}^n \to \mathbb{C}$ as

$$\mathcal{K}\phi(\boldsymbol{x}_i) = \phi \circ \mathbf{F}(\boldsymbol{x}_i) = \phi(\mathbf{F}(\boldsymbol{x}_i)) = \phi(\boldsymbol{y}_i), \tag{4}$$

yields a brand new scalar valued function that gives the value of $\phi$ *one-step ahead in the future* against the discrete dynamical system $(n, \mathcal{M}, \mathbf{F})$, where $n \in \mathbb{Z}$, $\mathcal{M} \subseteq \mathbb{R}^N$ and $\boldsymbol{x} \mapsto \mathbf{F}(\boldsymbol{x})$. With data sets recorded in matrix $\boldsymbol{X}$ and $\boldsymbol{X}^\bowtie$, the basic DMD based on singular value decomposition (SVD Trefethen & Bau (2022)) to provide the spectral-observables of $\mathcal{K}$ is given in Tu et al. (2014).

In the natural interest of determining the Koopman spectra-observables i.e. Koopman eigen-values $(\mu_k)$ and Koopman eigen-functionals $(\varphi_k)$, they are also accompanied by the Koopman modes $(\boldsymbol{\xi}_k)$ of a certain vector valued observable $\boldsymbol{g} : \mathcal{M} \to \mathbb{R}^{N_o}$, $(N_o \in \mathbb{N})$, which is refer as the *full state observable* given as $\boldsymbol{g}(\boldsymbol{x}) = \boldsymbol{x}$. Further, one can have a following decomposition in terms of the aforementioned the triple *eigen-values, eigen-functionals & modes* of the Koopman operator corresponding to the (unknown) dynamics $\boldsymbol{x} \mapsto \mathbf{F}(\boldsymbol{x})$: $\boldsymbol{x} = \sum_{k=1}^{N_k} \boldsymbol{\zeta}_k \varphi_k(\boldsymbol{x})$, $\mathbf{F}(\boldsymbol{x}) = \sum_{k=1}^{N_k} \mu_k \boldsymbol{\zeta}_k \varphi_k(\boldsymbol{x})$, where, supposing that $N_k$ is the number of tuples required for the re-construction of the system from the data of the dynamical system.

The eDMD is provided with the choice of scalar observables and for that let $\mathcal{F}$ be the appropriate choice of scalar observables (such as RKHS). To do this, let $\psi_k : \mathcal{M} \to \mathbb{R}$ for $k = 1, \ldots, N_k$ under the assumption that $\mathrm{span}(\mathcal{F}_{N_k}) \subset \mathcal{F}$. In particular, the space of scalar observables is approximated using $\{\psi_k\}_{k=1}^{N_k}$ functions then feature space is $\mathbb{R}^{N_k}$. Additionally, the *feature map $\boldsymbol{\psi}$* will be the 'stacked' column vector of entries $\{\psi_k\}$ given as $\boldsymbol{\psi}(\boldsymbol{x}) = \begin{bmatrix} \psi_1(\boldsymbol{x}) & \psi_2(\boldsymbol{x}) & \cdots & \psi_{N_k}(\boldsymbol{x}) \end{bmatrix}^\top$. With the feature space $\mathbb{R}^{N_k}$ and considering the value of any functions $\phi, \tilde{\phi} \in \mathcal{F}_{N_k}$, where (again) $\mathrm{span}\,\mathcal{F}_{N_k} \subset \mathbb{R}^{N_k}$, one can define the evaluation of both $\phi$ and $\tilde{\phi}$ against the inner product with certain coefficient vector $\boldsymbol{a}$ and $\tilde{\boldsymbol{a}}$ in $\mathbb{C}^{N_k}$:

$$\phi(\boldsymbol{x}) = \langle \boldsymbol{a}, \boldsymbol{\psi}(\boldsymbol{x}) \rangle_{\mathbb{R}^{N_k}} = \boldsymbol{\psi}(\boldsymbol{x})^\top \boldsymbol{a}, \quad \tilde{\phi}(\boldsymbol{x}) = \langle \tilde{\boldsymbol{a}}, \boldsymbol{\psi}(\boldsymbol{x}) \rangle_{\mathbb{R}^{N_k}} = \boldsymbol{\psi}(\boldsymbol{x})^\top \tilde{\boldsymbol{a}}.$$

The goal of the eDMD is to employ the pair of data-set of snapshots defined in equation 3 to generate the compactified[1] version of the Koopman operator denoted by $\boldsymbol{\mathcal{K}} \in \mathbb{R}^{N_k} \times \mathbb{R}^{N_k}$ for some given coefficients $\boldsymbol{a}$ and $\tilde{\boldsymbol{a}}$ such that $\mathfrak{r} = \left( \mathcal{K}\phi - \tilde{\phi} \right) \in \mathcal{F}$, is minimum. The eDMD algorithm is given as:

---

[1]finite rank representation of the infinite dimensional Koopman operator

---

**Algorithm 1** Extended-DMD algorithm Williams et al. (2015a)

Step 1 With the pair of data-set of snapshots as defined in equation 3, compute following:

$$\boldsymbol{\Psi_x} \triangleq \begin{bmatrix} \boldsymbol{\psi}(\boldsymbol{x}_1) \ \boldsymbol{\psi}(\boldsymbol{x}_2) \ \cdots \ \boldsymbol{\psi}(\boldsymbol{x}_M) \end{bmatrix}; \boldsymbol{\Psi_y} \triangleq \begin{bmatrix} \boldsymbol{\psi}(\boldsymbol{y}_1) \ \boldsymbol{\psi}(\boldsymbol{y}_2) \ \cdots \ \boldsymbol{\psi}(\boldsymbol{y}_M) \end{bmatrix} \in \mathbb{R}^{M \times N_k}.$$

Step 2 Compute $\boldsymbol{\mathfrak{G}} = \boldsymbol{\Psi_x}^\top \boldsymbol{\Psi_x}$ and $\mathbb{A} = \boldsymbol{\Psi_x}^\top \boldsymbol{\Psi_y}$.

Step 3 Determine pseudo-inverse of $\boldsymbol{\mathfrak{G}}$. Denote it by $\boldsymbol{\mathfrak{G}}^{(-)\mathfrak{p}}$.

Step 4 Determine $\mathcal{K}$ by $\mathcal{K} \triangleq \boldsymbol{\mathfrak{G}}^{(-)\mathfrak{p}} \mathbb{A}$.

---

In the spirit of SVD based DMD via Schmid (2010); Tu et al. (2014), the SVD of $\boldsymbol{\Psi_x}$ can be used to construct a matrix similar to $\mathcal{K}$; this is given as follows:

**Proposition 2.2.** *Let the SVD of $\boldsymbol{\Psi_x}$ be $\boldsymbol{\Psi_x} \triangleq \boldsymbol{Q\Sigma Z}^\top$, where $\boldsymbol{Q}$ and $\boldsymbol{\Sigma} \in \mathbb{R}^{M \times M}$ and $\boldsymbol{Z} \in \mathbb{R}^{N_k \times M}$. The pair of non-negative $\mu$ and $\hat{\boldsymbol{v}}$ are respective an eigenvalue and eigenvector of*

$$\hat{\mathcal{K}} \triangleq \left( \boldsymbol{\Sigma}^{(-)\mathfrak{p}} \boldsymbol{Q}^\top \right) \left( \boldsymbol{\Psi_y} \boldsymbol{\Psi_x}^\top \right) \left( \boldsymbol{Q} \boldsymbol{\Sigma}^{(-)\mathfrak{p}} \right) = \left( \boldsymbol{\Sigma}^{(-)\mathfrak{p}} \boldsymbol{Q}^\top \right) \hat{\mathbb{A}} \left( \boldsymbol{Q} \boldsymbol{\Sigma}^{(-)\mathfrak{p}} \right), \qquad (5)$$

*if and only if $\mu$ and $\boldsymbol{v} = \boldsymbol{Z}\hat{\boldsymbol{v}}$ are an eigen-value and eigen-vector of $\mathcal{K}$.*

## 3 PROPOSED WORK

The choice of kernel functions can dramatically change the performance of ML routine Singh (2024); Geifman et al. (2020b;a), specifically in those situation when shorter training time or limited and sparse data is available. The most common kernel function that arises from the class of radial basis function (cf. Fasshauer (2007)) used in ML and AI routine such as speech enhancement is the class of *Laplace Kernel* given as follows:

$$K_{\text{EXP}}^{1,\sigma} \triangleq K_{\text{EXP}}^{1,\sigma}(\boldsymbol{x}, \boldsymbol{z}) \coloneqq \exp\left( -\frac{\|\boldsymbol{x} - \boldsymbol{z}\|_2}{\sigma} \right) \qquad \text{LAPLACE KERNEL.}$$

Laplace Kernel plays a critical role to characterize Deep Neural Tangent Kernel Chen & Xu (2021) and finds remarkable application in various ML tasks Chen et al. (2021a); Ghojogh et al. (2021); Belkin et al. (2018); Geifman et al. (2020b); Genton (2001). We are motivated by the works Geifman et al. (2020b;a), where they used the Laplacian Kernels to execute their ML task in presence of either partial data or sparse data. This is so because, loosely speaking we share the common theme of predicting the data based on limited sparse information. Lap-KeDMD algorithm to predict ST-modes with irregular and sparse data snapshots is given as:

---

**Algorithm 2** Proposed Lap-KeDMD algorithm

Step 1 Construct limited data-set snapshots matrix which are collected in irregular and sparse manner.

Step 2 Compute Gram-Matrix $\boldsymbol{\mathfrak{G}} \coloneqq [\boldsymbol{\mathfrak{G}}]_{i \times j}$ and Interaction-Matrix $\boldsymbol{\mathcal{I}} \coloneqq [\boldsymbol{\mathcal{I}}]_{i \times j}$ by following:

$$[\boldsymbol{\mathfrak{G}}]_{i \times j} = K_{\text{EXP}}^{1,\sigma}(\boldsymbol{x}_i, \boldsymbol{x}_j), \qquad\qquad [\boldsymbol{\mathcal{I}}]_{i \times j} = K_{\text{EXP}}^{1,\sigma}(\boldsymbol{y}_i, \boldsymbol{x}_j).$$

Step 3 Determine the spectral observables of the Gram-matrix $\hat{\boldsymbol{\mathfrak{G}}}$, i.e. $\boldsymbol{Q}$ and $\boldsymbol{\Sigma}$.

Step 4 Construct $\hat{\mathcal{K}}$ via equation 5.

---

Reconstructed results from Step 4 in Algorithm 2 are yielded through the dominant Koopman modes calculated by the complex eigenvalues of the Gram Matrix whose real part's magnitude are comparatively larger helping in identifying the underlying patterns of the actual data of dynamical system. Dominant Koopman eigenvalues determined in such way is fundamentally important and thus plays a crucial role Williams et al. (2015b); Baddoo et al. (2022); Mezić (2021). Further, Koopman eigenvalues on an unit circle symbolizes the oscillatory mode of data, while on other hand, eigenvalues

lying inside an unit circle symbolizes decaying mode of data Kutz et al. (2016). We will compare the experimental results in Subsubsection 3.2 generated by the Laplacian Kernel and the GRBF Kernel. To collect the experimental results from the GRBF Kernel, we execute the workflow of Algorithm 2 by considering $[\boldsymbol{\mathfrak{G}}]_{i \times j} = K_{\text{EXP}}^{2,\sigma}(\boldsymbol{x}_i, \boldsymbol{x}_j)$, $[\boldsymbol{\mathcal{I}}]_{i \times j} = K_{\text{EXP}}^{2,\sigma}(\boldsymbol{y}_i, \boldsymbol{x}_j)$.

## 3.1 CLOSABILITY RESULTS

For KeDMD methods, as already first noted by Colbrook (2023) and then also by, Giannakis & Valva (2024) a challenge associated with RKHS techniques is that RKHS fails to exhibit invariance under the action of the Koopman operator. Further, Koopman operator invariance problem is actually tied with ST reconstruction of data matrix Bevanda et al. (2024) as well. The solution to this problem lies in choosing a suitable RKHSs where Koopman operators need to be closable Ikeda et al. (2022b). If in case closability criteria is not satisfied, this immediately will implicates that ST reconstruction is not possible. In the light of this, this section contains important theoretical results which leads up to the justification of closability of Koopman operators over the RKHS of Laplacian Kernel.

Throughout the paper, class of *observables functions* is captured by *holomorphic (entire) functions* (Markushevich, 2005, Chapter 9) mapping between $\mathbb{C}^n \to \mathbb{C}$. In order to systematic define the Laplacian Kernel as an $L^2-$measure, we consider an injective linear operator $\mathfrak{I}_-$ over $\mathbb{C}^n$ as $\mathfrak{I}_- \boldsymbol{z} = -\boldsymbol{I}_n \boldsymbol{z}$, where $\boldsymbol{I}_n$ is the identity matrix. Define the graph of $\mathfrak{I}_-$ as:

$$_-\mathfrak{Z}_+(\mathfrak{I}_-) := \{(\boldsymbol{z}, \mathfrak{I}_- \boldsymbol{z}) \in \mathbb{C}^n \times \mathbb{C}^n : \boldsymbol{z} \in \mathbb{C}^n\}$$
$$= \{(\boldsymbol{z}, -\boldsymbol{z}) \in \mathbb{C}^n \times \mathbb{C}^n : \boldsymbol{z} \in \mathbb{C}^n\} \triangleq {}_-\mathfrak{Z}_+. \quad (6)$$

Graph $_-\mathfrak{Z}_+$ introduced above will play a crucial role towards the end of paper in proving *closability* of Koopman operators. Laplacian Kernel as probability $L^2-$ measure $d\mu_{\sigma,1,\mathbb{C}^n}(\boldsymbol{z}) \, \forall \sigma > 0$ is

$$d\mu_{\sigma,1,\mathbb{C}^n}(\boldsymbol{z}) := \frac{1}{(2\pi\sigma^2)^n} K_{\text{EXP}}^{1,2\sigma}(-\mathfrak{Z}_+) dV(\boldsymbol{z}) = \frac{1}{(2\pi\sigma^2)^n} \exp\left(-\frac{\|\boldsymbol{z}\|_2}{\sigma}\right) dV(\boldsymbol{z}) \quad (7)$$

where $\|\boldsymbol{z}\|_2 = \sqrt{|z_1|^2 + \cdots + |z_n|^2}$ for $\boldsymbol{z} = (z_1, \ldots, z_n) \in \mathbb{C}^n$. Then, we can provide the inner product between two holomorphic functions $f : \mathbb{C}^n \to \mathbb{C}$ and $g : \mathbb{C}^n \to \mathbb{C}$ associated with this measure as

$$\langle f, g \rangle_{\sigma,1,\mathbb{C}^n} := \frac{1}{(2\pi\sigma^2)^n} \int_{\mathbb{C}^n} f(\boldsymbol{z}) \overline{g(\boldsymbol{z})} e^{-\frac{\|\boldsymbol{z}\|_2}{\sigma}} dV(\boldsymbol{z}). \quad (8)$$

Now, that we have inner product, we can have following as the Hilbert function space corresponding to the same.

$$H_{\sigma,1,\mathbb{C}^n} := \{\text{holomorphic function } f : \mathbb{C}^n \to \mathbb{C} : \|f\|_{\sigma,1,\mathbb{C}^n} < \infty\}. \quad (9)$$

Hilbert space $H_{\sigma,1,\mathbb{C}^n}$ equipped with the inner product equation 8 constitutes the RKHS (cf. Definition A.1) due to the existence of its orthonormal basis Theorem A.3 by the virtue of Theorem A.1.

**Theorem 3.1.** *For $\sigma > 0$, let $\boldsymbol{z} = (z_1, \ldots, z_n)$ and $\boldsymbol{w} = (w_1, \ldots, w_n)$ be in $\mathbb{C}^n$. Then the reproducing kernel for RKHS $H_{\sigma,1,\mathbb{C}^n}$ is*

$$K_{\boldsymbol{w}}^\sigma(\boldsymbol{z}) = K^\sigma(\boldsymbol{z}, \boldsymbol{w}) := \left(\sqrt{\frac{\langle \boldsymbol{z}, \boldsymbol{w} \rangle_{\mathbb{C}^n}}{\sigma^2}}\right)^{-1} \cdot \sinh\left(\sqrt{\frac{\langle \boldsymbol{z}, \boldsymbol{w} \rangle_{\mathbb{C}^n}}{\sigma^2}}\right) \quad (10)$$

*where $\langle \boldsymbol{z}, \boldsymbol{w} \rangle_{\mathbb{C}^n} = \boldsymbol{z} \overline{\boldsymbol{w}^\top} = \sum_{i=1}^n z_i \overline{w_i}$.*

Now that we have the RKHS $H_{\sigma,1,\mathbb{C}^n}$, we are ready to define the Koopman operators $\mathcal{K}_\varphi$ induced by holomorphic symbols $\varphi : \mathbb{C}^n \to \mathbb{C}^n$ over the RKHS $H_{\sigma,1,\mathbb{C}^n}$. This definition fundamentally helps in operator theoretic quantification over the RKHS $H_{\sigma,1,\mathbb{C}^n}$. It should be noted that, the way we exposed this definition is traditional follows from the setting of Cowen (1983); Carswell et al. (2003); Cowen Jr (2019); Hai et al. (2016); Hai & Khoi (2018); Hai & Rosenfeld (2021); Le (2014; 2017); Gonzalez et al. (2024).

**Definition 3.1.** *Let $\varphi : \mathbb{C}^n \to \mathbb{C}^n$ be a holomorphic function in which every coordinate function of it are holomorphic functions from $\mathbb{C}^n \to \mathbb{C}$. Then, the Koopman operator induced by $\varphi$ is denoted by $\mathcal{K}_\varphi : \mathcal{D}(\mathcal{K}_\varphi) \subset H_{\sigma,1,\mathbb{C}^n} \to H_{\sigma,1,\mathbb{C}^n}$ and is the linear operator defined by*

$$\mathcal{K}_\varphi(f) := f \circ \varphi.$$

*The domain of $\mathcal{K}_\varphi$ is $\mathcal{D}(\mathcal{K}_\varphi)$ given as $\mathcal{D}(\mathcal{K}_\varphi) := \{f \in H_{\sigma,1,\mathbb{C}^n} : f \circ \varphi \in H_{\sigma,1,\mathbb{C}^n}\}$.*

Detailed Koopman operator theoretic quantification in terms of *boundedness* and *compactness* along with its proofs are provided in Subsubsection A.5 and Subsubsection A.6. Now, we will discuss the closability of the Koopman operators over the RKHS $H_{\sigma,1,\mathbb{C}^n}$ and the RKHS of the GRBF Kernels. We reviewed the closability of linear (unbounded) operators in Appendix B. Our workflow of showing the closability of the Koopman operators over the RKHS $H_{\sigma,1,\mathbb{C}^n}$ is based on Lemma B.1.

**Theorem 3.2.** *Let $\mathcal{K}_\varphi$ acts boundedly over the RKHS $H_{\sigma,1,\mathbb{C}^n}$ with $\varphi(z) = Az + b$. Define a subsequence of $_-\mathfrak{Z}_+$ by $_-\mathfrak{Z}_{+,N}$ as*

$$_-\mathfrak{Z}_{+,N} := \left\{ (z_N, -z_N) \in \mathbb{C}^n \times \mathbb{C}^n : \lim_{N \to \infty} \|z_N\|_2 = 0 \text{ where } z \in \mathbb{C}^n \right\}. \tag{11}$$

*There exists a sequence $\mathfrak{K}_N$ inside the RKHS $H_{\sigma,1,\mathbb{C}^n}$ as defined $\mathfrak{K}_N := \|z_N\|_2 K^\sigma(_-\mathfrak{Z}_{+,N})$ such that $\lim_{N \to \infty} \mathfrak{K}_N = 0$. Also, $\mathcal{K}_\varphi$ is closable over the RKHS $H_{\sigma,1,\mathbb{C}^n}$ if $b \equiv 0$, implying $\varphi$ is linear in $\mathbb{C}^n$, that is $\varphi(z) = Az$. Let $\varphi_A(z) := Az$, then $\lim_{N \to \infty} \mathcal{K}_{\varphi_A} \mathfrak{K}_N = 0$.*

*Proof.* See Subsubsection B.1 $\qquad\qquad$ □

In order to discuss the closability of Koopman operators on the GRBF Kernel, we recall it as follows subsequently followed by its RKHS as well.

$$K_{\text{EXP}}^{2,\sigma} \triangleq K_{\text{EXP}}^{2,\sigma}(x, z) := \exp\left( -\frac{\|x - z\|_2^2}{\sigma} \right) \qquad\qquad \text{GRBF KERNEL.}$$

Norm for the GRBF Kernel Steinwart et al. (2006); Steinwart & Christmann (2008) is

$$\|f\|_\sigma^2 := \frac{2^n \sigma^{2n}}{\pi^n} \int_{\mathbb{C}^n} |f(z)|^2 e^{\sigma^2 \sum_{i=1}^n (z_i - \bar{z}_i)^2} dV(z), \tag{12}$$

where $dV(z)$ is the usual Lebesgue volume measure on $\mathbb{C}^n \equiv \mathbb{R}^{2n}$. RKHS for $K_{\text{EXP}}^{2,\sigma}(x, z)$ is:

$$H_\sigma := \{f : \mathbb{C}^n \to \mathbb{C} : f \text{ is holomorphic and } \|f\|_\sigma < \infty\}. \tag{13}$$

Following theorem leverages the bounded Koopman operators over the RKHS $H_\sigma$ defined in equation 13 whose result is borrowed from (Gonzalez et al., 2024, Corollary 1, Page 8).

**Theorem 3.3.** *Let $\mathcal{K}_\varphi$ be bounded on $H_\sigma$, then $\varphi = Az + b$, where $A \in \mathbb{C}^{n \times n}$ and $b \in \mathbb{C}^n$. Recall the domain $_-\mathfrak{Z}_+$ defined in equation 6 and the subspace $_-\mathfrak{Z}_{+,N}$ given in equation 11. Then $\lim_{N \to 0} \|z_N\|_2 K_{\text{EXP}}^{2,\sigma}(_-\mathfrak{Z}_{+,N}) = 0$ but $z_N K_{\text{EXP}}^{2,\sigma}(_-\mathfrak{Z}_{+,N}) \notin H_\sigma$. Since the sequence $\left\{ z K_{\text{EXP}}^{2,\sigma}(_-\mathfrak{Z}_{+,N}) \right\}_N \notin H_\sigma$, hence $\mathcal{K}_\varphi$ is not closable over the RKHS $H_\sigma$.*

*Proof.* See Subsubsection B.2. $\qquad\qquad$ □

## 3.2 RESULTS

We consider a number of important *benchmark experiments* to investigate the performance of the proposed Lap-KeDMD algorithm under limited irregular and sparse data snapshots. These experiments find wide-range applications in real-world setting and have been used by various practitioners Bevanda et al. (2024); Colbrook (2023); Colbrook & Townsend (2024); Baddoo et al. (2022); Bagheri (2013); Williams et al. (2015a;b); Rosenfeld et al. (2022); Gonzalez et al. (2024).

However, previous work relied on the collection of spatio-temporal data in uniform time sampled manner. We consider the same benchmark experiments by irregularly sampling sparse data with the same goal of determining ST modes via the compacted Koopman operators achieved over the RKHS of Laplacian kernel (Theorem A.19) and GRBF kernels (Gonzalez et al. (2024)). Explanations are in the Appendix.

Additionally we consider a real-world spatio-temporal data collected from Seattle freeway traffic speed sensors (experiment 4). This dataset has innate irregularity and sparsity due to the placement of sensors. Moreover, there is no known underlying dynamical system generating this dataset. This makes the reconstruction of this data set extremely challenging. In every experiment, we compare

the spatio-temporal reconstruction using our proposed Laplacian Kernel $K_{\text{EXP}}^{1,1}$ (Algorithm 2) with the state-of-the art GRBF Kernel $K_{\text{EXP}}^{2,1}$ (replacing $K_{\text{EXP}}^{1,1}$ by $K_{\text{EXP}}^{2,1}$ in Algorithm 2).

Table 1: Governing equations of experiments (if available).

| EXPERIMENT | GOVERNING EQUATION(S) |
| --- | --- |
| 1. | $\dfrac{\partial u}{\partial t} + u\dfrac{\partial u}{\partial x} = \nu\dfrac{\partial^2 u}{\partial x^2}$ |
| 2. | $\dfrac{\partial}{\partial t}\mathbf{u}(x,y,t) + \mathbf{u}(x,y,t)\cdot\nabla\mathbf{u}(x,y,t) + \nabla p(x,y,t) - \dfrac{1}{\text{RE}}\nabla^2\mathbf{u}(x,y,t) = 0.$ |
| 3. | $\dot{\boldsymbol{x}} = \boldsymbol{y},\ \dot{\boldsymbol{y}} = -0.5\boldsymbol{y} + \boldsymbol{x} - \boldsymbol{x}^3.$ |
| 4. | n/a |

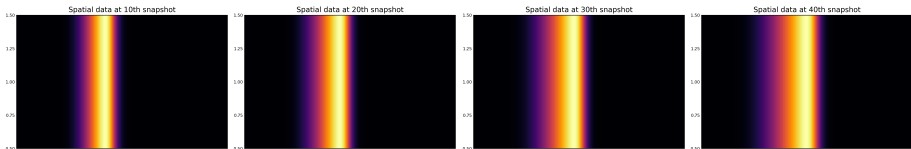

Figure 1: EXPERIMENT 1: *Nonlinear Burger's Equation.* Spatial evolution of 256 spatial values over the time span of $[0, 10]$ sec. Presented are the spatial data snapshots collected at the time stamp $10^{\text{th}}$, $20^{\text{th}}$, $30^{\text{th}}$ and $40^{\text{th}}$.

Table 2: EXPERIMENT 1: *Nonlinear Burger's Equation.* ST reconstruction through dominant Koopman modes via $K_{\text{EXP}}^{1,1}$ and $K_{\text{EXP}}^{2,1}$ with irregular and sparse 40 snapshots out of actual 100 snapshots.

| # KOOPMAN MODE | GROUND TRUTH | IRREGULAR & SPARSE | RECON. VIA LAP | RECON. VIA GRBF | KOOPMAN EIGENVALUES (LAP, GRBF) |
| --- | --- | --- | --- | --- | --- |
| # $39^{\text{th}}$ | | | | | $0.99 + 0i, 0.99 + 0i$ |

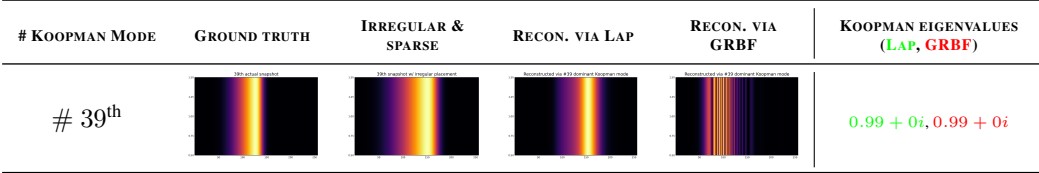

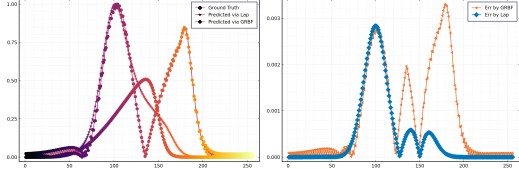

Figure 2: EXPERIMENT 1: *Nonlinear Burger's Equation.* Spatial reconstruction result along with ground truth for spatial data present at $39^{\text{th}}$ snapshot by both $K_{\text{EXP}}^{1,1}$ and $K_{\text{EXP}}^{2,1}$ (*left*). Absolute error plots for the same as well (*right*), where performance by $K_{\text{EXP}}^{2,1}$ is poor as compared to $K_{\text{EXP}}^{1,1}$.

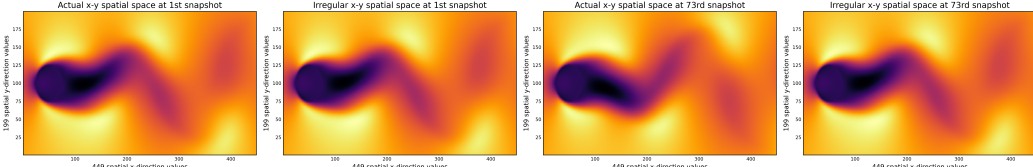

Figure 3: EXPERIMENT 2: *Fluid flow across cylinder.* A sample of $1^{\text{st}}$ and $73^{\text{rd}}$ snapshots out of available 151 snapshots for $2-$D spatial plots of $x \times y \in \mathbb{R}^{449} \times \mathbb{R}^{199}$ spatial sensors values evolving with respect to time uniformly over time space $[1, 151]$ sec (scaled) in $1^{\text{st}}$ and $3^{\text{rd}}$ entries. Irregular and and sparse spatial $2-$D plots of the same are given in $2^{\text{nd}}$ and $4^{\text{th}}$ entries respectively. Few of entries of uniform time sampling recorded by 151 time sensors (scaling) are $[1, 2, 3, \ldots, 151]^{\top}$ sec while on the other hand, the irregular version of the same is given as $[151, 148, 144, 9, 6, \ldots, 94, 98, 97, 103]^{\top}$ sec.

Table 3: EXPERIMENT 2: *Fluid flow across cylinder.* ST reconstruction through dominant Koopman modes via $K_{\text{EXP}}^{1,1}$ and $K_{\text{EXP}}^{2,1}$ with irregular and sparse 100 snapshots out of total 151 snapshots. Labeling for both spatial directions $x$ and $y$ is same from Figure 3.

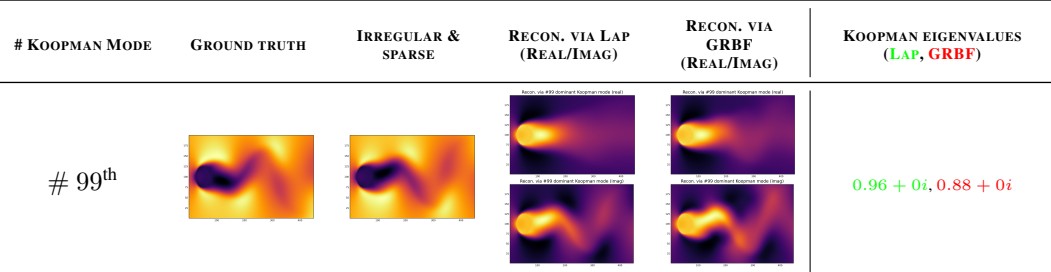

| # KOOPMAN MODE | GROUND TRUTH | IRREGULAR & SPARSE | RECON. VIA LAP (REAL/IMAG) | RECON. VIA GRBF (REAL/IMAG) | KOOPMAN EIGENVALUES (LAP, GRBF) |
|---|---|---|---|---|---|
| # $99^{\text{th}}$ | | | | | $0.96 + 0i, 0.88 + 0i$ |

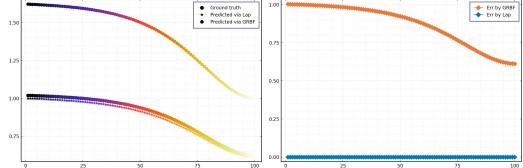

Figure 4: EXPERIMENT 2: *Fluid flow across cylinder.* Spatial reconstruction results along with ground truth for $2-$D spatial data present at the $99^{\text{th}}$ snapshot by both $K_{\text{EXP}}^{1,1}$ and $K_{\text{EXP}}^{2,1}$ (*left*). Absolute error plots for the same as well (*right*).

Governing $2-$D equations for chaotic Duffing Oscillator is already given in entry 3 in Table 1. Dataset for this experiment build by an initial condition given as $[-1.8760, 1.7868]^{\top}$. This experiment actually belongs to the natural setting of phase-portrait reconstruction problem which we have adopted the same to test proposed Lap-KeDMD algorithm in terms of spatial-modes reconstruction.

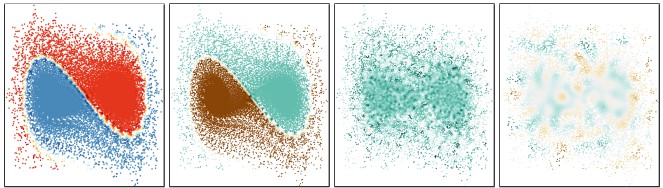

Figure 5: EXPERIMENT 3: *Duffing Oscillator.* Chaotic Duffing Oscillator with actual full dataset (ground truth in first entry), irregular and sparse of the same (second entry), prediction via $K_{\text{EXP}}^{1,1}$ (third entry) and prediction via $K_{\text{EXP}}^{2,1}$ (fourth entry) based on provided 35000 trajectories values.

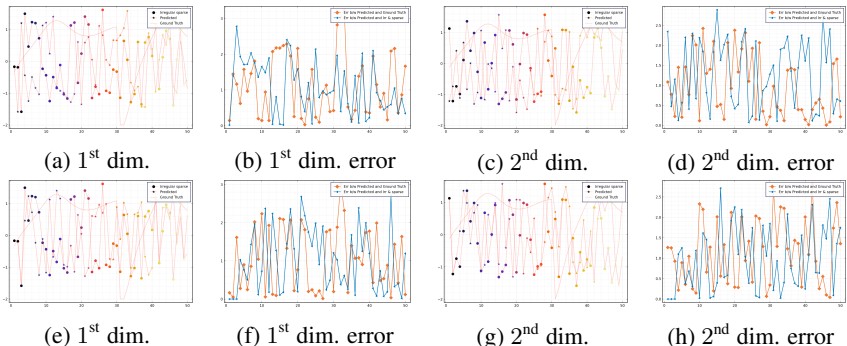

| (a) $1^{st}$ dim. | (b) $1^{st}$ dim. error | (c) $2^{nd}$ dim. | (d) $2^{nd}$ dim. error |
| (e) $1^{st}$ dim. | (f) $1^{st}$ dim. error | (g) $2^{nd}$ dim. | (h) $2^{nd}$ dim. error |

Figure 6: EXPERIMENT 3: *Duffing Oscillator*. State space reconstruction in both dimensions used for phase portraits reconstruction in entries third and fourth in Figure 5. First state space is $\boldsymbol{x} \in \mathbb{R}^{35000}$ followed by second space $\boldsymbol{y} \in \mathbb{R}^{35000}$. Reconstruction of these $\boldsymbol{x}$ and $\boldsymbol{y}$ by $K_{\mathrm{EXP}}^{1,1}$ are given in Figure 6a and Figure 6c. On the other hand, reconstruction of these $\boldsymbol{x}$ and $\boldsymbol{y}$ by $K_{\mathrm{EXP}}^{1,1}$ are given in Figure 6e and Figure 6g. When we reconstruct and compare the complete phase portraits in third and fourth entries in Figure 5, we see that the result by $K_{\mathrm{EXP}}^{1,1}$ outperforms the $K_{\mathrm{EXP}}^{2,1}$.

Table 4: EXPERIMENT 4: *Seattle I-5 Freeway Traffic Speed data*. ST reconstruction through dominant Koopman modes via $K_{\mathrm{EXP}}^{1,1}$ and $K_{\mathrm{EXP}}^{2,1}$. For this experiment, real and imaginary parts of eigenvalues delivered by $K_{\mathrm{EXP}}^{2,1}$ are in the order of $O(10^{-q})$, $q \ggg 1$ due to ill-condition of Gram Matrix.

| # KOOPMAN MODE | GROUND TRUTH | RECON. VIA LAP | RECON. VIA GRBF | KOOPMAN EIGENVALUES (LAP, GRBF) |
|---|---|---|---|---|
| # $23^{th}$ | | | | $-0.28 - 0.46i, 0 + 0i$ |

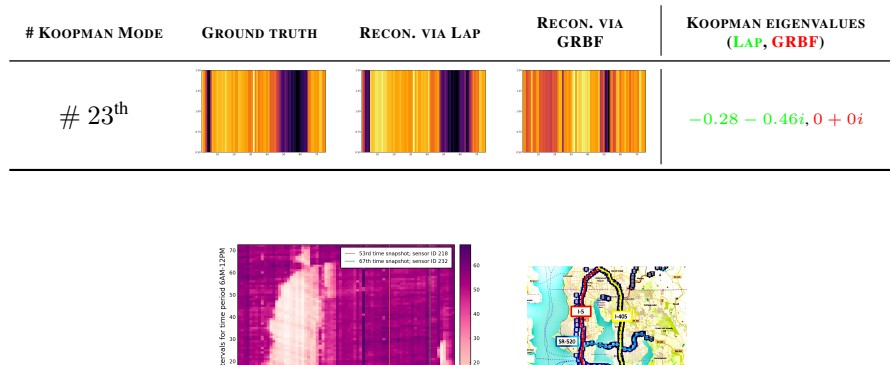

Figure 7: EXPERIMENT 4: *Seattle I-5 Freeway Traffic Speed data*. (*Left*) For the Seattle I-5 freeway traffic speed data, there are 75 loop detectors recording vehicle's speed in a 72 time intervals frame over a rush-hour period 6:00 am — 12:00 pm; follow Cui et al. (2018; 2019) for more details. These 75 loop detectors present on freeway have sensor IDs as 166 till 240 in sequentially manner. (*Right*) Map of Seattle freeway from Cui et al. (2018; 2019) showing their inductive loop detector locations where each blue icon indicate loop detectors at every traffic sensor present at every milepost.

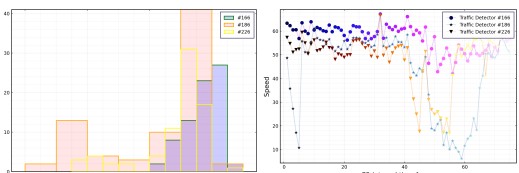

Figure 8: EXPERIMENT 4: *Seattle I-5 Freeway Traffic Speed data*. Histogram (speed vs. histogram bins) of speed data on Seattle I-5 freeway traffic speed data collected by sensors with loop detector IDs $166^{th}$, $186^{th}$ and $226^{th}$ (*left*). Plot of Seattle I-5 freeway speed collected on these sensors (*right*).

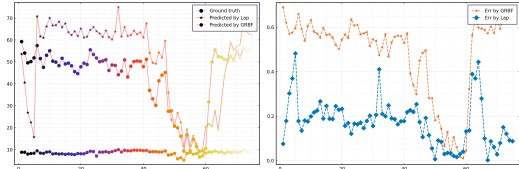

Figure 9: EXPERIMENT 4: *Seattle I-5 Freeway Traffic Speed data*. Spatio-temporal speed reconstruction result along with the ground truth for speed data present at the Seattle I-5 freeway traffic sensor ID $218^{\text{th}}$ by both $K_{\text{EXP}}^{1,1}$ and $K_{\text{EXP}}^{2,1}$ (*left*). Absolute error plots for the same as well (*right*). Speed data collected at the Seattle I-5 freeway sensor ID $218^{\text{th}}$ corresponds to $53^{\text{rd}}$ time snapshot.

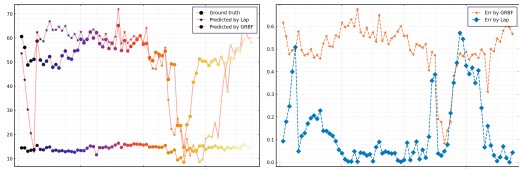

Figure 10: EXPERIMENT 4: *Seattle I-5 Freeway Traffic Speed data*. Spatio-temporal speed reconstruction result along with the ground truth for speed data present at the Seattle I-5 freeway sensor ID $232^{\text{th}}$ snapshot by both $K_{\text{EXP}}^{1,1}$ and $K_{\text{EXP}}^{2,1}$ (*left*). Absolute error plots for the same as well (*right*). Speed data collected at the Seattle I-5 freeway sensor ID $232^{\text{nd}}$ corresponds to $67^{\text{th}}$ time snapshot.

## 4 CONCLUSION

We presented Lap-KeDMD to discover ST modes from a given limited spatio-temporal dataset and compared the results in contrast to that of GRBF Kernel using spectral observables of Koopman operators. We quantify bounded closable and compact Koopman operators on RKHS $H_{\sigma,1,\mathbb{C}^n}$. Also, main reason why we fail to achieve the closability of the Koopman operators over the GRBF Kernel is because of the inner-product for the function space corresponding to the GRBF Kernel. In particular, the measure present in the norm for the function space in equation 12 is unable to make the $L^2-$integration finite. Since the *closability* of Koopman operator is directly linked with the ST-mode reconstruction, we see that reconstruction for the experiments via Laplacian Kernel are of higher quality in terms of information richness and data measure predictions as opposed to that of GRBF Kernel. We already anticipate such results due to the closability of the Koopman operators over the the RKHS of Laplacian Kernel as opposed to that of the GRBF Kernel. For future directions, it will be interesting to investigate the workflow of the Lap-KeDMD algorithm to recover the ST-modes of data if we have partial information of the dynamical system.

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

## A APPENDIX

### A.1 REPRODUCING KERNEL HILBERT SPACE

The definition of reproducing kernel Hilbert space (RKHS) is given as follows:

**Definition A.1.** *Let $X = \emptyset$ and $(H, \langle \cdot, \cdot \rangle_H)$ be the Hilbert function space over $X$.*

