# OpenReview forum: "Sparse spatio temporal reconstruction with Closable Kernel Space"
_ICLR.cc/2025/Conference — ICLR 2025 Conference Withdrawn Submission_

### Official Review · Reviewer_ZEcg · 2024-10-29

**Soundness:** 1
**Presentation:** 1
**Contribution:** 2
**Rating:** 3
**Confidence:** 3

**Summary:**

This paper investigates a kernelized (extended) dynamic mode decomposition (eDMD) for reconstruction of spatiotemporal data under the Koopman operator framework. The authors present _Laplace kernel eDMD_, which uses the Laplace kernel in the kernelized formulation of eDMD. They show that compared to the popular Gaussian RBF kernel, the Koopman operator acting on the RKHS of the hyperbolic sine kernel with linear observables is closable, whereas on the RKHS of the RBF kernel, it is not. This suggests the ability / inability of Laplacian / Gaussian kernel to reconstruct spatiotemporal modes from limited data.

**Strengths:**

The investigation of closability of Koopman operators over RKHSs and its implications on the corresponding eDMD algorithm using the respective reproducing kernel is an interesting perspective (although I am not sure if I have fully understood it). The main results are furthermore supported by various experiments comparing the kernelized eDMD using the Laplace vs Gaussian kernel, showing that the former leads to better reconstruction results.

**Weaknesses:**

Overall, I find the writing to be difficult to understand. The logical flow of the paper is hard to follow and I struggle to see how the main theoretical result of the paper tells us  about the proposed algorithm outside from an educated guess (see also questions below). I wish the authors made it clearer how the closabilty of Koopman operators over an RKHS tells us about the ability of the algorithm to reconstruct spatiotemporal data. At present, I find this to be confusing since the kernel in the proposed algorithm is given by the Laplace kernel, but the RKHS used in the main result is _not_ the one corresponding to the Laplace kernel but instead the _hyperbolic sine kernel_. A further explanation of how these two kernels are related would be useful for the understanding of the main result, in addition to why the closability result allows spatiotemporal reconstruction.

In terms of the experiments, insufficient details are provided, with the only explanations available in the captions of the various figures. This makes it difficult for us to understand the purpose and outcomes of the various experiments presented. Overall, more care should be taken in the presentation of the results, highlighting (1) the purpose of each experiment, (2) basic experimental configurations, and (3) what the results tell us about Laplace vs Gaussian kernel. In addition, the labels in the figures are too small to read without zooming the screen. This should be made much bigger.

**Questions:**

There are some points regarding the main results of the paper where I am unsure and hope the authors could please clarify. I believe these are critical and impact the validity of the theoretical claims made in the paper.
- In the Lap-KeDMD algorithm, the Laplace kernel is used to generate the gram and interaction matrices. However, the main theoretical result (Theorem 3.2) is stated in terms of a different hyperbolic sine kernel (10), which is the reproducing kernel of a Laplace-weighted $L^2$-space $H_{\sigma, 1, \mathbb{C}^n}$. I don't understand the relation between the Laplace kernel and this Laplace-weighted $L^2$-space. In particular, I hope the authors can clarify further why Theorem 3.2, which use the hyperbolic sine kernel, says anything about Algorithm 2, which use the Laplace kernel.
- I am unsure why Theorem 3.2 establishes closability over the RKHS $H_{\sigma, 1, \mathbb{C}^n}$ since they only show the property $  \lim_{n\rightarrow\infty} x_n = 0, Tx_n = y_n \Rightarrow \lim_{n\rightarrow\infty} y_n = 0$ over _a single sequence_ $x_n := \|z_n\|_2K^\sigma(z_n, -z_n)$. Whereas Lemma B.1 states that this should be shown for all such sequences $x_n$ to establish closability.
- I am not convinced with the proof of Theorem 3.3, since the authors only show that _an upper bound_ to $\|z_N\|_2 \exp(-4\|z\|_2^2/\sigma)$ is unbounded. This says nothing about $\|z_N\|_2 \exp(-4\|z\|_2^2/\sigma)$ being unbounded.

---

> ### Author Response · Authors · 2024-11-26
> **Clarification**
>
> The authors would like to apologize to the reviewer and withdraw the paper; but also would like to thanks as well for providing useful comments. The investigation of closability of the Koopman operators over the underlying Hilbert spaces (RKHS or etc.) is indeed a new field and is quite a desirable feat.
>
> * The particular way of utilizing the Laplacian Kernel and embedding it into the Laplacian measure in $L^2-$measure theoretic sense is deeply inspired by the RKHS study of the Gaussian measure $\exp(-|z|^2)dA(z)$ which generates the reproducing kernel, called as \emph{exponential dot-product kernel} given as $\exp(zw^\top)$ \cite[Chapter 2]{zhu2012analysis}. The aforementioned Hilbert space is referred to as \emph{Bergmann-Seigel-Fock Spaces} \cite{janson1987hankel} or simply Fock space and keeps a direct connection to the initial value problems such as heat equation and etc. as well
>
> * Doing the construction of RKHS in the above described way is not new as other mathematician also have made notable attempts like this in \cite{colorado1963lectures,baez2014introduction,zhu2012analysis} for the Gaussian measure. Note that composing Gaussian measure with quantity $\|x-y\|$ yields \emph{Gaussian Radial Basis Function Kernel}. Recently such study was also augmented by incorporating Generalized Gaussian Radial Basis Function Kernel as well, refer \cite{singh2024machine}. Hence, the resulting function space is now rich enough to initiate the investigation of almost every operators, possible unbounded as well.
>
> * The reviewer questions our argument on the choice of certain sequence to satisfy the closability condition of Koopman operators over the RKHS of the Laplacian Kernel $H_{\sigma,1,\mathbb{C}^n}$. We would like to let the reviewer know that our sequence was arbitrarily chosen satisfying a  certain norm condition, that is, $\lim_{N\to\infty}\|\bm{z}\|_2=0$, resulting from the graph of $_-\mathfrak{Z}_{+,N}$ (eq. 11). As this sequence happens to be arbitrary, this should be satisfying the general condition of closability criteria. We also would like to let the reviewer know that we provide a simple interpretation Lemma B.1 in in-text line number 1394, which authors think might be helpful.
>
> * The concern here is this that the particular quantity `$\|\bm{z}_N\|_2\exp\left(-4\frac{\|\bm{z}_N\|_2^2}{\sigma}\right)$` fails to exist in the RKHS corresponding to the GRBF Kernel let alone that sequence will provide the closability of the Koopman operator. However, we fail to see such scenario in the RKHS of Laplacian Kernel $H_{\sigma,1,\mathbf{C}^n}$ that we have uniquely constructed.

---

### Official Review · Reviewer_mxe2 · 2024-11-01

**Soundness:** 1
**Presentation:** 1
**Contribution:** 1
**Rating:** 1
**Confidence:** 5

**Summary:**

This is a nice attempt by AI to submit a fake technical paper to one of the top ML/AI conferences. It successfully fooled me into reading the first few pages with quite some interest until I found it makes no sense coming to the techniques. Then I downloaded the supplement, only to find a 118.8MB python notebook (no bother to open) and an appendix full of nonsense.

**Strengths:**

It contains an interesting topic. Submitted is a paper that appears technically sound.

**Weaknesses:**

* Most part does not make sense. Even the term "closable" is not explained. The whole paper is full of grammatical errors. It surely does not explain how the "fundamental issue of ST reconstruction on irregular grids" is solved by Koopman operator.

* Waste of time for reviewers.

**Questions:**

Too many to ask.

---

> ### Author Response · Authors · 2024-11-26
> **Maybe the review was done by AI**
>
> We are very concerned with the tone and the confidence with which this review is written.
>
>
> Most part does not make sense. ---> What part? Especially, What math did not make sense?
>
> Even the term "closable" is not explained ---> (Definition B.3, Page 26)
>
> The whole paper is full of grammatical errors---> Duly noted.
>
> It surely does not explain how the "fundamental issue of ST reconstruction on irregular grids" is solved by Koopman operator ---> When did we claim this? We are trying to say the non-closability of Koopman operator leads to a fundamental issue in the ST reconstruction. We do apologize for the lack of clarity in this regard.
>
>
> Waste of time for reviewers ---> Maybe our paper was too mathematical in the area of expertise for the reviewer or maybe we should rewrite it with better exposition; hoping to get some feedback on what it is so that we are better prepared next time. We apologize for wasting reviewer's time; we know the reviewers have too much responsibilities in their hand.

---

### Official Review · Reviewer_LPmL · 2024-11-10

**Soundness:** 1
**Presentation:** 2
**Contribution:** 2
**Rating:** 3
**Confidence:** 3

**Summary:**

The authors propose to use the Laplace kernel for approximating Koopman operators on RKHSs. They insist that with the Laplace kernel, the Koopman operators are closable.

**Strengths:**

Analyzing Koopman operators theoretically is an important and interesting topic.

**Weaknesses:**

I mainly have following concerns:
**Readability**
This paper is hard to follow since the notation is not consistent and there are undefined notions. For example,
- In Definition 2.2, what is $\mathbb{X}$?
- In Definition 2.3, what is $n$? the authors say $n$ is the time, but $n$ is also the dimension of the state space.
- In line 144, what is the definition of $\xi_k$? I couldn't figure out why $x$ is assumed to be represented using the Koopman eigenvectors and $F(x)$ is in such a form.
- In line 149, what is $\mathcal{F}_{N_k}$?
- In line 160, what is the relationship between $\phi$ and $\tilde\{\phi\}$?
- In Proposition 2.2, what is (-)p? Does this mean psudo inverse?

**Validity of the proposed method**
The authors propose to use the Laplace kernel for KDMD (Algorithm 2). However, in Section 3.1, they define a Hilbert space equipped with the inner product (9). I could not figure out the relationship between the RKHS associated with the Laplace kernel and the Hilbert space defined in Section 3.1.

**Correctness of the result**
In Theorem 3.3, $\mathcal{K}\_{\phi}$ is assumed to be bounded on the whole Hilbert space, but the authors conclude $\mathcal{K}_{\phi}$ is not closable. I could not figure out why a bounded operator defined on the Hilbert space is not closable. In addition, in my understanding, Koopman operators on an RKHS is closed. Thus, I'm concerned with the correctness of the result. Indeed, in the proof of Theorem 3.3, I could not figure out the relationship between constructing a sequence converging to $0$ and the Koopman operator.

**Questions:**

Could you provide the detailed proof of Theorem 3.3?

---

> ### Author Response · Authors · 2024-11-26
> **Clarification**
>
> We thank the reviewer for their time. We will withdraw our paper, however we are concerned about the skepticism raised on the validity of our mathematical work. We wan to clarify this.
>
>
> We want to point out a main issue. Koopman operator in general is not bounded and we are not sure if we pointed this anywhere. Indeed this depends on the system at hand. The reviewer also acknowledges the general concern on operator theoretic topic where a bounded operator on RKHS is closed. The koopman operator being just closed  would not be able to provide or exhibit invariance under the action of the Koopman operator \cite{colbrook2023multiverse}; for that one needs the Koopman operator (being specific here) to be **closable**. Competitive conceptual scopes, nowadays, in the data-driven analysis community are facing this crucial challenge in quantifying the closability condition of the Koopman operator. Fortunately, despite of this tough situation, we are able to do so in this paper.
>
> * The reviewer raises the concern on the closability of the Koopman operators `$\mathcal{K}_{\phi}$` over the RKHS `$H_{\sigma,1,\mathbb{C}^n}$` by asking about the detailed proof of Theorem 3.3. The detailed proof is present in the Appendix section B.1. Note that we are looking for an arbitrary sequence converging to $0$ that can suffice the closability criteria. Further, the same sequence which do exist in the RKHS corresponding to that of GRBF Kernel fails to let the corresponding Koopman operator closable.
>
>
> * The notation $\mathbb{X}$ corresponds to the standard (spatial) function space, commonly as $\mathbb{C}$ in single variable sense. Follow \cite{mezic2020spectrum} for more details on this.
>
> * The authors would like to apologize for this particular confusion. Here, $n$ specifically corresponds to \emph{only} the dynamical system discretized $n-$times.
>
> * We follow the presented details from the standard paper of \emph{Extended Dynamic Mode Decomposition by Williams et. al. \cite[Eq. 2 and Eq.3]{williams2015data}}. These details are necessary since our Lap-KeDMD algorithm relies on their work
>
> * $\mathcal{F}_{N_k}$ is the subspace spanned by $N_k$ chosen Koopman eigenfunctions corresponding to the data. Here, we encourage the reviewer to follow the aforementioned paper by Williams et. al. \cite[Page 1312 Second Paragraph]{williams2015data}. We further, would like to let know the reviewer that we take the liberty to slightly use different notations as compared to that of what used in \cite{williams2015data} however, we assure that we meant the same conceptual aspect of the concerned topic.
>
> * The relationship between $\phi$ and $\Tilde{\phi}$ is already given in line number 156, with corresponding explanation preceding this line number (line number 154, 155 specifically). We request the reviewer to follow this, if needed!
>
> * The reviewer is right and we agree with that. We apologize for not specifying it before in our manuscript.

---

### Note · Authors · 2024-11-26

**Comment:**

We have decided to withdraw the paper and address reviewer concerns. We think our work was more mathematical and needed more clarity with regards to the spatiotemporal reconstruction problem.  We found some issues from the reviewers puzzling.  Koopman operator in general is not bounded and we are not sure if we pointed this anywhere. Indeed this depends on the system at hand. The reviewer also acknowledges the general concern on operator theoretic topic where a bounded operator on RKHS is closed. The koopman operator being just closed  would not be able to provide or exhibit invariance under the action of the Koopman operator \cite{colbrook2023multiverse}; for that one needs the Koopman operator (being specific here) to be **closable**. Thus, we think our most important mathematical contribution was not realized by the reviewers. Maybe this is due to our writing.
However we want to comment on the validity of the math which some reviewers have commented on. In particular, one reviewer (mxe2) was very confident that the paper was written by AI and did not even check the math, we are very concerned for this kind of review to be a part of ICLR review process and question whether this reviewer is capable of reviewing mathematical details of this paper. We suggest ICLR chair to read some of the comments made by this reviewer -- in my experience there is no place of such a review in any community.

We have collected some responses to valid concerns and responded more in detail to the respective reviewers.



### Reply to mxe2
The reviewer finds this paper misleading in terms of our commitment towards the understanding of complex issues of dynamical system and portraying it as an ML/AI paper. The authors of this paper would like to sincerely apologize for this inconvenience, but we are completely aghast with the absolute certainty with which the author has dismissed the paper. We don't think the reviewer went through the math. It is completely an insult to say we haven't defined closability (our main contribution in this paper) (Definition B.3, Page 26). We hope this helps the smooth navigation of reviewer for this paper.

**Withdrawal Confirmation:**

I have read and agree with the venue's withdrawal policy on behalf of myself and my co-authors.